# The reliability of medical illness reporting in a randomized clinical trial

**Rachelle Morgenstern[1], Avi Reichenberg[2], Benjamin Kummer[1], Nathalie Jette[3], Mark J. Kupersmith** [4]*

**1** Icahn School of Medicine at Mount Sinai, New York, New York, United States of America,
**2** Icahn School of Medicine at Mount Sinai, Icahn School of Medicine at Mount Sinai, Pscyhiatry and Environemental Medicine and Public Health, New York, New York, United States of America,
**3** Departments of Clinical Neurosciences and Community Health Sciences, O'Brien Institute for Public Health, University of Calgary, Calgary, Canada, **4** Icahn School of Medicine at Mount Sinai, Ophthalmology and Neurosurgery, New York, New York, United States of America

* mark.kupersmith@mountsinai.org

## Abstract

### Background/Objective

Reported medical disorders from population surveys, medical records, and clinical trials, may not be accurate and methods are needed to improve confirmation. We report the accuracy of reported prevalence of medical disorders in a clinical trial and comparison with potential verification methods.

### Methods

We report the prevalence of 11 medical disorders, utilizing prospectively collected data from 729 participants in an eight-country multicenter clinical treatment trial on non-arteritic anterior ischemic optic neuropathy (NAION). We chose disorders where the medical history was potentially verifiable. We determined the prevalence using four methods: Method (M)1: Participant and medical health record reporting; M2: Physical examination, clinical tests; M3: Medication indications; M4: Combining M2 and M3. We estimated concordance between M1 and the other methods using Cohen's kappa (K) statistic.

### Results

Prevalence of the medical disorders based on M1 were lower than for either M2 or M3, depending on the disorder, and consistentlly lower for M4. For M1 and M4, moderate concordance (K ≥ 0.50) was observed only for psychiatric disorders (K = 0.52) and prior NAION (K=0.67). The prevalence and concordance for M1 and M4 for anemia, hypertension, diabetes and psychiatric disease were the only disorders that differed between females and males. For all methods, the prevalence varied widely

**Data availability statement:** Others who seek access to the dataset used in this study have to contact Quark Pharmaceutical, Inc for a data sharing agreement. Quark Pharmaceutical, Inc allowed MJK (me) access to the data for academic use. The binding agreement with the company prevents distribution of the data for use as the database contains proprietary information that could be used by competitor drug companies. All data queries should be directed to dan@cafaroconsulting.com.

**Funding:** This study was funded by: The New York Eye and Ear Infirmary Foundation, New York, N.Y. (MJK); National Eye Institute of National institute of Health EY032522 (MJK); Research to Prevent Blindness, Inc., New York, NY unrestricted grant to the Department of Ophthalmology (MJK); Shulman Family NAION Fund at Icahn School of Medicine at Mount Sinai (MJK), National Institute of Health P30 EY003790 (AR).

**Competing interests:** NO authors have competing interests.

**Abbreviations:** M, method; ALB, Albumin; ALP, alkaline phosphatase; ALT, alanine transaminase; AST, aspartate aminotransferase; BILI, bilirubin; BMI, body mass index; BP, blood pressure; BUN, blood urea nitrogen; CHOL, cholesterol; CI, confidence interval; CR, creatinine; CV, cardiovascular; DM, diabetes mellitus; ECG, electrocardiogram; HB, hemoglobin; HCT, hematocrit; HLD, hyperlipidemia; HTN, hypertension; IHD, ischemic heart disease; ICD, ischemic cerebrovascular disease; LDH, lactate dehydrogenase; NAION, Nonarteritic anterior ischemic optic neuropathy; PRT, protein; RBC, red blood cells; SPB, systolic blood pressure;TG, triglycerides; U/L, units per liter.

across countries. Concordance for M1 and M4 varied and moderate concordance occurred for psychiatric disorders and prior NAION.

## Conclusion

Even with prospective, rigorously collected data, medical histories do not reliably identify all medical disorders. Adding the results of physical examination, laboratory tests, and medications increases the accuracy of reporting. This strategy could be adapted for clinical trials and electronic medical record disease-prevalence data mining.

## Introduction

Healthcare providers, clinical trialists, and electronic healthcare record (EHR) system users collect patient and study participant self-reported medical histories, systemic medical problem lists, laboratory test results and medications. These databases are used by healthcare providers, clinical trialists and data scientists to facilitate patient care and determine eligibility or stratification in clinical trials. Specific medical illness reporting is also the focus of government agencies and organizations with inferred prevalence and metrics about relevant diseases using a variety of methods including surveys. Currently, there is no universal method to verify the veracity and completeness of these data in clinical trial databases, surveys, and EHRs in the United States. Yet, decisions are made daily based on these information sources.

Prospectively collected data is considered best for avoiding the bias that occurs in retrospective record review and should therefore be most useful to determine concordance or prevalence of disease or comorbidities for specific illnesses. But medical illness reporting is still dependent on the degree that each component is linked in each database. A prospective study conducted at an inner-city emergency department reported of 114 patients, only 71 (62%) provided accurate histories concerning major health issues such as cardiac, pulmonary, neurological, and sickle cell when compared with the information in existing medical records.[1] Another prospective study performed in an emergency department showed that medication-use was the variable most prone to discrepancy. Comparing pharmacist-determination and patient self-reporting in 138 patients revealed approximately 20% had at least one discrepancy for a major medication.[2]

Studies that target reporting reliability show increased accuracy if patients are queried on a single medical disorder or if patients are told in advance that the study purpose is to test the accuracy of responses. When patients were asked about an orthopedic issue, patient accuracy increased to 90%.[3] A prospective study in a cancer population showed a 90% concordance for patient self-reported diseases and recorded medical histories when the patients were informed the issue was the evaluation of their self-reporting.[4] These reports suggest that a regimented data collection at study enrollment and the nature of individuals who enroll in a

clinical trial could provide more accurate identification of their medical illnesses. Participants in clinical trials are generally recruited at a particular stage of an illness and usually have uniformly collected medical information during the trial. Thus, a clinical trial database, which is structured comprehensively, should provide an excellent source to explore the veracity of disease prevalence data and develop methods to improve accuracy.[5]

As we explored a database for a recently completed clinical trial of non-arteritic anterior ischemic optic neuropathy (NAION), we noted discrepancies in the medical history prevalence data when compared with the medications, examination and laboratory data. This database provided an opportunity to investigate potential discrepancies in the reporting of the frequency of major illnesses in this defined cohort. We hypothesized that the study ancillary data such as medications, physical examination findings, and laboratory test results collected at screening and enrollment for this clinical trial would verify medical record and patient-reported medical illnesses, in this study population.

## Methods

### Study cohort

We utilized the reported medical history, physical examination, blood and other clinical test results, and medications from Quark Pharmaceutic Company (trial sponsor), Qrk 207 NAION study database collected at the screening and enrollment visits to explore the veracity of reported medical illnesses.[6] The clinical trial was registered at ClinicalTrials.gov with Identifier: NCT02341560 and conducted after Institutional Review Board (IRB) approval according to the tenets of the Declaration of Helsinki. The trial began 02/24/16 and was completed 07/01/19. The data was received and accessed for research purposes starting 01/04/21. The study enrolled 729 subjects, at 80 study sites in eight countries, with acute NAION who met entry criteria (detailed in supplement), some of which led to not including an estimated 12% of patients who had exclusionary medical disorders. The analysis of these collected data for the purpose of the current study required no additional IRB approval according to the Icahn School of Medicine at Mount Sinai IRB as the data were de-identified. Quark Pharmaceuticals provided a complete secure de-identified database for all study data collected, including a complete standardized medical and ophthalmic history, physical examination, blood and urine laboratory tests, electrocardiogram (ECG), ocular exam findings, and medications. All participants were 50–80 years old and had acute unilateral NAION for no longer than 14 days at enrollment.

### Data collection and categorization

The information in the Qrk 207 NAION study was collected and entered by certified study coordinators at each site. Medical illness reporting, by participants or taken from site EHRs (at some but not all sites) or both, clinical tests, physical examination findings (performed by site investigators) and medications were recorded in accordance with the study protocol. In this report, we focused on the occurrence of 11 medical disorders of interest (Table 1), for which potential verifying confirmation was available in the database. The evidence included medications, fasting basic blood laboratory test results, other clinical tests and physical examinations. The categories of reported medical disorders were mostly obvious, e.g., diabetes mellitus, but some were grouped into one entity, e.g., cerebrovascular and peripheral venous and arterial diseases were termed vascular disorders. Cardiac disorders, most of which were ischemic heart disease, were separated from vascular disorders (see supplement for complete list). We also were aware that the exclusion criteria for the clinical trial would limit the frequency of active or severe systemic diseases, but this would not alter the purpose of this report – to investigate methods of recording medical illness.

### Laboratory testing

We standardized the blood laboratory values using the normal values and ranges for Mount Sinai Beth Israel Laboratory standard operating procedures for 2021. We categorized values that were above or below reference ranges as abnormally

**Table 1. Blood laboratory results, physical exam findings, and other tests limits used in the methods to verify the medical disorders.**

| Medical Disorder | Labs/Tests/Physical Exam Findings Assessed and Abnormal Criteria* |
|---|---|
| **Anemia** | Hb [7]<br>*Low male < 14.0 g/dL*<br>*Low female < 12.0 g/dL* |
| | Hct [8]<br>*Low male < 41%*<br>*Low female < 36%* |
| | RBC<br>*Low male < 4.5 (cells/mcL)*<br>*Low female < 3.7 (cells/mcL)* |
| **CV disorders** | ECG *(categorized as abnormal)* |
| | HR *< 60 bpm/* HR *>100 bpm* [9] |
| | Physical exam |
| **DM** | Blood glucose > *126 mg*/dL [10] |
| **Hepatic disorders** (abnormal ≥ 2 labs) [11,12] | Serum ALB < *3.5 g/dL* |
| | Serum ALP > *126 U/L* |
| | Serum ALT > *55 U/L* |
| | Serum AST ≥ *36 U/L* |
| | Serum BILI ≥ *0.9 mg/dL* |
| | Serum LDH > *200 U/L* |
| **HLD** | Serum total CHOL ≥ *240 mg/dL* [13] |
| | Serum TG ≥ *151 mg/dL* [14] |
| **ystemic HTN** | BP measurement<br>*SPB ≥ 130 or DBP >80* [15] |
| **Vascular disorders** (including ICD) | Physical exam |
| **Kidney disorders** | Serum Cr [16]<br>*High Male > 1.4 mg/dL*<br>*High Female > 1.2 mg/dL* |
| | BUN/Cr > 20 or < 10< 1 |
| **Obesity** | BMI ≥ *30* |
| **Prior NAION in fellow eye** | Visual field and fundus photo evaluation[6] |
| **Psychiatric disorders** | N/A |

*Outside of blood test reference range noted in methods and following recommendations from medical specialty organizations.

high or low, respectively. We used laboratory or physical exam or test result criteria recommended in the literature or by specialty societies or organizations to categorize each participant as having each disorder (Table 1).

## Medication indications

We expanded the sponsor list of indications for each medication, noted in the database, using the Anatomical Therapeutic Chemical codes [17], the FDA and National Library of Medicine [18] databases for additional indications. We grouped medications as indicators of having specific medical illnesses, recognizing that some have multiple potential indications. Thus, specific medications were categorized for more than one clinical indication or disorder.

## Statistical analysis

We measured the frequency and concordance for diagnoses using four ascertainment methods: M1 - Participant and health record reported medical history alone; M2 - A combination of physical examination findings and clinical or laboratory tests results; M3 - Medications grouped by indications and; M4 - Combining all the results of methods 2 and 3.

The prevalence of reported medical disorders and the determinants or verifiable measures of medical illnesses, including laboratory measures, medications, and clinical exam findings, were calculated as frequencies with proportions (%). The determinants were stratified and compared across country of residence and sex using Chi-square tests, and when appropriate Fisher exact tests. We used Cohen's kappa (K) statistic to analyze the relative observed agreement corrected for the probability of chance agreement between reported medical history for the 11 medical illnesses and the determinants used in methods 2, 3 and 4. The K statistic ranges from -1–1, such that a negative measurement indicated less than chance agreement, zero specified chance agreement, and a positive measurement indicated greater than chance agreement.[19] K statistics with 95% confidence intervals were calculated across country of residence and sex. We also examined potential modification by sex and by country using Chi-square statistic tests to assess whether the stratum levels of the K statistics were equal.[20] For analyses stratified by country, we included 725 participants, excluding the four participants from Singapore. For all other analyses, we included all 729 participants. For each of the 11 diseases and their corollary determinants, we conducted power analyses. Assuming a null hypothesis of K = 0.81, which indicates substantial agreement, and an alpha of 0.05, we found that a sample size ranging from a minimum of 101 to a maximum of 275 produced 98% power. When we assumed a null hypothesis of K = 0.6, indicating moderate agreement, a sample size ranging from a minimum of 197 to a maximum of 551, produced 98% power. Consequently, our sample size appeared to be sufficient to test our hypotheses. Statistical analyses were performed using SAS software version 9.4 (SAS Institute Inc., Cary, NC).[21]

Funding Source: None of the funding sources had a role in the study design, conduct, and reporting.

## Results

The study cohort included 500 males (mean age 61 ± 7.8 years) and 229 females (mean age 62.1 ± 7.4 years). The race composition for the study participants was 570 White, 149 Asian, and three other races from eight countries.

### Primary analyses

By history alone, the reported prevalence of medical disorders for all study participants was variable (Table 2). None of the reported disorders occurred in more than 30% of participants, except for systemic hypertension and vascular disorders. However, laboratory and other tests and the physical examination (M2) or for the medications (M3) were abnormal in more than 24% of participants for cardiac disorders, diabetes, hyperlipidemia, systemic hypertension, and obesity (supplement Table 1). Medical history reporting in eight disorders was uncommonly unassociated with at least one of the other determinants of disease (supplement Table 2). Remarkably, for three medical disorders that affected at least 100 participants and could be verified by laboratory tests or physical examination (M2) or medications (M3), medical reporting with no verifying determinants occurred in 11% for hyperlipidemia, 14% for high blood pressure, and 19% for prior NAION in the fellow eye. The agreement between the reported medical disorder (M1) and the M2 determinants, in general, were minimal or poor for all medical disorders except for prior fellow eye NAION. The medical disorders with the lowest amount of agreement with medications (M3) were kidney disease [-0.003 (-0.007, 0.002)], obesity [-0.02 (-0.05, 0.01)] and cardiac disorders [0.07 (0.04, 0.11)]. The medical disorder history with the lowest agreement with the laboratory test results were hepatic disorders [0.01 (-0.03, 0.04)], hyperlipidemia [0.07 (0.03, 0.11)] and anemia [0.06 (0.01, 0.11)]. The medical disorders with the lowest agreement with combining all three verification methods (M4) were anemia [0.06 (0.01, 0.11)], cardiac disorders [0.07 (0.05, 0.09)], hyperlipidemia [0.06 (0.04, 0.09)], and kidney disease [0.05 (0.01, 0.1)]. However, there were some instances of moderate agreement between a reported medical disorder with disorder-specific medication use including diabetes [0.59 (0.52, 0.66)], psychiatric disorders [0.52 (0.44, 0.61)], and systemic hypertension [0.39 (0.32, 0.46)]. The combination of disorder-specific medications, laboratory test, and physical examination evaluations (M4) yielded a frequency that was markedly greater than each reported medical disorder alone

**Table 2. Prevalence of reported medical history[5] and agreement between the reported medical history and other methods used for medical illness verification for entire cohort.**

| Method/Characteristic** | Preva-lence (%) | Kappa statistic, (95% CI) | Positive for disease by report and/or determinant(s) (%) |
|---|---|---|---|
| M1 *History of anemia* | 1.2 | | 21.1 |
| M3 Iron supplement | 1 | 0.24 (-0.04, 0.52) | |
| M2 Abnormally low lab(s) (HCT, HB, RBC)*** | 20.2 | 0.06 (0.01, 0.11) | |
| M4 Iron and/or abnormally low lab(s) (HCT, HB, RBC) | 21 | 0.08 (0.03, 0.13) | |
| M1 *History of cardiac disorder* | 13.7 | | 79.8 |
| M3 Cardiac medications | 64.8 | 0.07 (0.04, 0.11) | |
| M2 Abnormal ECG | 21.4 | 0.36 (0.22, 0.5) | |
| M2 Abnormal HR | 24.5 | 0.04 (-0.07, 0.14) | |
| M3 Abnormal ECG and/or HR | 41.6 | 0.15 (0.07. 0.23) | |
| M4 Cardiac medications and/or abnormal ECG and/or HR | 79.4 | 0.07 (0.05, 0.09) | |
| M1 *History of diabetes mellitus* | 15.1 | | 57.9 |
| M3 Diabetes medication | 24.6 | 0.59 (0.52, 0.66) | |
| M2 Abnormally high glucose | 43.9 | 0.05 (0.02, 0.08) | |
| M4 Diabetes medication and/or abnormally high glucose | 57.6 | 0.22 (0.18, 0.26) | |
| M1 *History of hepatic disorder* | 0.6 | | 47.6 |
| M2+ Abnormally high labs (ALT, AST, ALP, BILI, LDH, PRT) | 14.1 | 0.01 (-0.03, 0.04) | |
| M1 *History of hyperlipidemia* | 15.2 | | 80.4 |
| M3 Anti-HLD medication | 42 | 0.13 (0.07, 0.19) | |
| M2 Abnormally high lab(s) (TG, CHOL) | 64.5 | 0.07 (0.03, 0.11) | |
| M4 Anti-HLD medication and/or abnormal lab(s) (TG, CHOL) | 79.4 | 0.06 (0.04, 0.09) | |
| M1 *History of systemic hypertension* | 30.3 | | 64.5 |
| M3 Anti-hypertension medications | 37 | 0.39 (0.32, 0.46) | |
| M2 Abnormally high blood pressure | 40 | 0.19 (0.1, 0.27) | |
| M4 Anti-HTN medications and/or abnormally high BP | 60.4 | 0.29 (0.24, 0.35) | |
| M1 *History of vascular disorders (including 7 ICD)* | 31.3 | | 71.2 |
| M3 Vascular disorder medications | 65.2 | 0.17 (0.12, 0.23) | |
| M1 *History of kidney disorder* | 1.2 | | 20.6 |
| M3 Kidney medication | 0.1 | -0.003 (-0.007, 0.002) | |
| M2 Abnormally high lab(s) (CR, BUN) | 11.5 | 0.18 (0.08, 0.28) | |
| M4 Kidney medication and/or abnormally high lab(s) (CR, BUN) | 20.5 | 0.05 (0.01, 0.1) | |
| M1 *History of obesity* | 20.7 | 0.05 (0.01, 0.1) | |
| M3 Obesity medications | 17.3 | -0.02 (-0.05, 0.01) | |
| M2 BMI > 30 | 32.3 | 0.08 (0.03, 0.12) | |
| M4 Obesity medications and/or BMI > 30 | 43.9 | 0.04 (0.02, 0.07) | |
| M1 *History of prior NAION* | 21 | | 28.7 |
| M2+ Physician diagnosed prior NAION | 24.7 | 0.67 (0.61, 0.73) | |
| M1 *History of psychiatric disorder* | 11.7 | | 20 |
| M3+ Psychiatric medications | 16.7 | 0.52 (0.44, 0.61) | |

+ also used for M4.

*Determinants include laboratory measures, medications, and/or physical exams;

**Characteristics are presented as frequency (percent);

***N (%) of patients' labs and clinical exams exclude those who took medication for the disease/disorder observed.

### Secondary analyses

In general, the prevalence of reported medical disorders (M1, previously reported[6]), laboratory test abnormalities and abnormal physical findings (M2), and medications (M3) were similar across sex for most characteristics (Table 3). Females and males had similar degree of agreement for test and exam abnormalities and medications (M4) with reported cardiac disorder [0.1 (0.06, 0.12) for males and 0.1 (0.05, 0.14) for females], diabetes mellitus [0.6 (0.52, 0.69) for males and 0.53 (0.38, 0.68) for females], systemic hypertension [0.27 (0.21, 0.33) for males and 0.35 (0.24, 0.46) for females]. Moderate concordance for medications (M3) and psychiatric disorder was similar for males [0.57 (0.45, 0.68) and females [0.45 (0.31, 0.58)]. Except for prior NAION in the fellow eye, the reported medical disorders did not have moderate or strong agreement with the laboratory findings or physical exam findings stratified by sex for all disorders.

The prevalence of reported medical disorders (M1), laboratory abnormalities and abnormal physical exam findings (M2) and medications (M3) varied widely among the seven countries with enough participants to analyze (supplement Table 3). There was a striking difference in the reporting of psychiatric disorders and use of medication to treat these disorders, with no participants in China with either, and one participant (4.4%) in Italy taking medication, while in the USA, 15% had a reported history and 23% took medication for a psychiatric disorder. The prevalence of anemia and renal dysfunction were too low to compare among countries.

The agreement between determinants M2 and M1 was fair to poor across all countries, except for prior fellow eye NAION in Australia, Israel, and the USA (Table 4). Concordance between medications (M3) and M1 was moderate to high across all countries, except in Israel, for diabetes, for systemic hypertension only in China and India, and for psychiatric disorders only in Australia and the USA. Agreement was moderate with disorder-specific medication or laboratory test results and physical findings (M4) for diabetes in China and Italy and systemic hypertension in China; but for all other disorders agreement was generally poor. Anemia, hepatic and renal disorders were too infrequent for analysis by country.

### Key points

**Question:** Are medical illnesses reported in clinical trials verifiable and accurate?

**Findings**: Analysis of the records of a clinical trial shows significant under-reporting of 11 major medical disorders. Verification methods, using laboratory tests results, physical examination findings, and medication indications show study participants have much greater prevalence of these illnesses.

**Meaning**: Verifiable objective methods can be used to improve the accuracy of medical disorder prevalence in all medical records.

### Discussion

Our study results show that self-reporting or medical history information collected from an EHR alone, even when collected in a clinical trial, is an unreliable method to determine the prevalence of major medical disorders. This finding has major implications for clinical decision making as well as research, as under reporting of illnesses is common. The reduced reporting can also slow or decrease recruiting to new clinical trials that depend on extracting research-quality phenotypes from the EHR. It also suggests that some attempts at personalized medicine may be biased; and research based on such data, which includes large genetic databases linked to EHR or self-reported data [22,23], may potentially lead to false findings, bias and wrong treatment recommendations. We were surprised that even when rigorously collected data in a clinical trial were compared with three objective verification methods, the observed reporting on common medical disorders was deficient. Our methods included diagnosis-determination based on the medication indications or physical examination findings and laboratory tests collected at study entry. The third approach, combining the results for the latter two methods, substantially increased identification of medical disorders. Our study is novel in that we used a database prospectively collected at a uniform time point prior to administering a study intervention. The trial collected the data utilizing a standardized protocol to evaluate medical illnesses (inclusive of psychiatric disorders) as well as for a

**Table 3. Prevalence of reported medical history[5] and methods for illness verification (determinants*) according to sex.**

| Method/Characteristic** | Prevalence, N (%) | | P-value | Kappa Statistic, (95% CI) | | P-value |
|---|---|---|---|---|---|---|
| | Male N = 500 | Female N = 229 | | Male N = 500 | Female N = 229 | |
| M1 *History of anemia* | 5 (1) | 4 (1.7) | NS | | | |
| M3 Iron supplement | 3 (0.6) | 4 (1.7) | NS | 0.24 (-0.15, 0.64) | 0.24 (-0.16, 0.64) | NS |
| M2 Abnormally low lab(s) (HCT, HB, RBC)*** | 118 (23.7) | 21 (9.3) | <.0001 | 0.05 (0.003, 0.1) | 0.15 (-0.05, 0.34) | NS |
| M4 Iron and/or abnormally low lab(s) (HCT, HB, RBC) | 121 (24.2) | 25 (10.9) | <.0001 | 0.06 (0.01, 0.1) | 0.22 (0.003, 0.43) | NS |
| M1 *History of cardiac disorder* | 71 (14.2) | 29 (12.7) | NS | | | |
| M3 Cardiac medications | 325 (65) | 147 (64.2) | NS | 0.07 (0.02, 0.11) | 0.09 (0.03, 0.15) | NS |
| M2 Abnormal ECG | 41 (23.4) | 14 (17.1) | NS | 0.39 (0.22, 0.55) | 0.28 (0.005, 0.55) | NS |
| M2 Abnormal HR | 21 (12) | 4 (4.9) | .02 | 0.07 (-0.10, 0.25) | -0.05 (-0.09, -0.02) | NS |
| M3 Abnormal ECG and/or HR | 198 (39.6) | 72 (31.4) | .03 | 0.31 (0.18, 0.45) | 0.22 (-0.01, 0.46) | NS |
| M4 Cardiac medications and/or abnormal ECG and/or HR | 379 (75.8) | 164 (71.6) | NS | 0.1 (0.06, 0.12) | 0.1 (0.05, 0.14) | NS |
| M1 *History of diabetes mellitus* | 83 (16.6) | 27 (11.8) | NS | | | |
| M3 Diabetes medication | 136 (27.2) | 43 (18.8) | .01 | 0.6 (0.52, 0.69) | 0.53 (0.38, 0.68) | NS |
| M2 Abnormally high glucose | 52 (14.3) | 65 (7.5) | .02 | 0.2 (0.07, 0.34) | 0.06 (-0.13, 0.24) | NS |
| M4 Diabetes medication and/or abnormally high glucose | 188 (37.6) | 57 (24.9) | .001 | 0.5 (0.41, 0.56) | 0.43 (0.3, 0.6) | NS |
| M1 *History of hepatic disorder* | 3 (0.6) | 1 (0.4) | NS | | | |
| M2+ Abnormally high labs (ALT, AST, ALP, BILI, LDH, PRT) | 76 (15.2) | 27 (11.8) | NS | 0.01 (-0.03,0.06) | -0.01 (-0.03, 0.01) | NS |
| M1 *History of hyperlipidemia* | 82 (16.4) | 29 (12.7) | NS | | | |
| M3 Anti-HLD medication | 217 (43.4) | 89 (38.9) | NS | 0.11 (0.04, 0.18) | 0.16 (0.06, 0.27) | NS |
| M2 Abnormally high lab(s) (TG, CHOL) | 129 (46.1) | 65 (46.8) | NS | 0.13 (0.04, 0.21) | 0.1 (0.01, 0.19) | NS |
| M4 Anti-HLD medication and/or abnormal lab(s) (TG, CHOL) | 346 (69.2) | 154 (67.2) | NS | 0.13 (0.05, 0.21) | 0.1 (0.01, 0.2) | NS |
| M1 *History of systemic hypertension* | 150 (30) | 71 (31) | NS | | | |
| M3 Anti-hypertension medications | 195 (39) | 75 (32.8) | NS | 0.4 (0.32, 0.48) | 0.36 (0.23, 0.49) | NS |
| M2 Abnormally high blood pressure | 129 (42.3) | 41 (26.6) | .001 | 0.17 (0.08, 0.26) | 0.26 (0.09, 0.43) | NS |
| M4 Anti-HTN medications and/or abnormally high BP | 324 (64.8) | 116 (50.7) | .0003 | 0.27 (0.21, 0.33) | 0.35 (0.24, 0.46) | NS |
| M1 *History of vascular disorders (including ICD)* | 18 (3.6) | 6 (2.6) | NS | | | |
| M3 Vascular disorder medications | 328 (65.6) | 147 (64.2) | NS | 0.03 (0.01, 0.05) | 0.03 (0.01, 0.05) | NS |
| M1 *History of kidney disorder* | 6 (1.2) | 3 (1.3) | NS | | | |
| M3 Kidney medication | 1 (0.2) | 0 (0) | NS | -0.003 (-0.01, 0.002) | **** | **** |
| M2 Abnormally high lab(s) (CR, BUN) | 92 (18.6) | 57 (24.9) | NS | 0.08 (0.01, 0.15) 7) | 0.01 (-0.05, 0.06) | NS |
| M4 Kidney medication and/or abnormally high lab(s) (CR, BUN) | 93 (18.7) | 57 (24.9) | NS | 0.08 (0.01, 0.15) | 0.01 (-0.05, 0.06) | NS |
| M1 *History of obesity* | 5 (1) | 7 (3.1) | NS | | | |
| M3 Obesity medications | 96 (19.2) | 30 (13.1) | .04 | 0.001 (-0.03, 0.04) | -0.05 (-0.09, -0.02) | .03 |
| M2 BMI ≥ 30 | 130 (32.4) | 64 (32.2) | NS | 0.04 (0.002, 0.08) | 0.14 (0.05, 0.24) | NS |
| M4 Obesity medications and/or BMI ≥ 30 | 226 (45.2) | 94 (41) | NS | 0.02 (0.003, 0.05) | 0.09 (0.03, 0.15) | NS |
| M1 *History of prior NAION* | 114 (22.8) | 39 (17) | NS | | | |
| M2+ Physician diagnosed prior NAION | 138 (27.6) | 42 (18.3) | .01 | 0.65 (0.57, 0.73) | 0.72 (0.6, 0.84) | NS |
| M1 *History of psychiatric disorder* | 40 (8) | 45 (19.7) | <.0001 | | | |
| M3+ Psychiatric medications | 65 (13) | 57 (24.9) | <.0001 | 0.57 (0.45, 0.68) | 0.45 (0.31, 0.58) | NS |

+also used for M4.

*Determinants include laboratory measures, medications, and/or physical exams;

**Characteristics are presented as frequency (percent);

***N (%) of patients' labs and clinical exams exclude those who took medication for the disease/disorder observed.

**Table 4. Agreement between reported medical disorder and methods of verification (determinants\*) of each medical illness according to country of residence.**

| Determinant (M)/ Medical disorder (M1) | Kappa Statistic, (95% CI) | | | | | | | P-Value |
|---|---|---|---|---|---|---|---|---|
| | **AUS**<br>**N = 27** | **CHN**<br>**N = 34** | **DEU**<br>**N = 23** | **IND**<br>**N = 111** | **ISR**<br>**N = 47** | **ITA**<br>**N = 23** | **USA**<br>**N = 460** | |
| Iron supplement (M3)/ Anemia (M1) | 0 (0, 0) | ** | ** | 0 (0, 0) | ** | ** | 0.28 (-0.04, 0.59) | ** |
| Abnormally low lab(s) (HCT, HB, RBC) (M2)/ Anemia (M1) | 0 (0, 0) | 0 (0, 0) | 0 (0, 0) | 0.02 (-0.02, 0.07) | 0 (0, 0) | 0 (0, 0) | 0.12 (0.02, 0.22) | NS |
| Iron and/or abnormally low lab(s) (HCT, HB, RBC) (M4)/ Anemia (M1) | 0 (0, 0) | 0 (0, 0) | 0 (0, 0) | 0.02 (-0.02, 0.07) | 0 (0, 0) | 0 (0, 0) | 0.15 (0.05, 0.25) | .03 |
| Cardiac medication (M3)/ Cardiac disorder (M1) | 0.07 (-0.06, 0.2) | 0.09 (-0.08, 0.27) | 0.2 (-0.12, 0.51) | 0.04 (-0.08, 0.16) | 0.04 (-0.01, 0.09) | 0.28 (0.02, 0.53) | 0.04 (-0.01, 0.08) | NS |
| Abnormal ECG (M2)/ Cardiac disorder (M1) | 0 (0, 0) | 0 (0, 0) | 1 (1, 1) | 0.32 (-0.04, 0.69) | ** | ** | 0.32 (0.16, 0.48) | NS |
| Abnormal HR (M2)/ Cardiac disorder (M1) | 0 (0, 0) | 0 (0, 0) | 0 (0, 0) | -0.02 (-0.05, 0.01) | ** | ** | 0.03 (-0.16, 0.22) | NS |
| Abnormal ECG and/or HR (M2)/ Cardiac disorder (M1) | 0 (0, 0) | 0 (0, 0) | 1 (1, 1) | 0.29 (-0.05, 0.63) | ** | ** | 0.26 (0.13, 0.39) | NS |
| Cardiac medication and/or abnormal ECG and/or HR/ Cardiac disorder (M1) | 0.04 (-0.04, 0.13) | 0.07 (-0.06, 0.19) | 0.32 (0.05, 0.6) | 0.1 (-0.004, 0.2) | 0.04 (-0.006, 0.09) | 0.28 (0.02, 0.53) | 0.06 (0.04, 0.08) | NS |
| Diabetes medication (M3)/ Diabetes mellitus (M1) | 0.65 (0.02, 1) | 0.66 (0.38, 0.93) | 1 (1, 1) | 0.61 (0.46, 0.75) | 0.36 (0.09, 0.63) | 1 (1, 1) | 0.55 (0.45, 0.65) | NS |
| Abnormally high glucose (M2)/ Diabetes mellitus (M1) | 0 (0, 0) | 0.33 (-0.25, 0.91) | ** | 0.17 (-0.04, 0.39) | 0 (0, 0) | 0 (0, 0) | 0.17 (0.02, 0.32) | NS |
| Diabetes medication and/or abnormally high glucose (M4)/ Diabetes mellitus (M1) | 0.47 (-0.13, 1) | 0.62 (.35,.89) | 1 (1, 1) | 0.45 (0.32, 0.58) | 0.22 (0.03, 0.42) | 0.78 (0.36, 1) | 0.46 (0.37, 0.55) | NS |
| Abnormally high lab (ALT, AST, ALP, BILI, LDH) (M2)+/ Hepatic disorder (M1) | 0 (0, 0) | 0 (0, 0) | 0 (0, 0) | 0 (0, 0) | -0.04 (-0.1, 0.02) | -0.07 (-0.18, 0.04) | 0.04 (-0.04, 0.12) | NS |
| Anti-hyperlipidemia medication (M3)/ Hyperlipidemia (M1) | -0.07 (-0.2, 0.06) | 0.43 (-0.03, 0.9) | 0.09 (-0.36, 0.55) | 0.14 (-0.09, 0.36) | -0.002 (-0.21, 0.2) | 0 (0, 0) | 0.12 (0.05, 0.18) | NS |
| Abnormally high lab(s) (TG, CHOL) (M2)/ Hyperlipidemia (M1) | -0.1 (-0.26, 0.06) | 0.06 (-0.2, 0.32) | 0.23 (-0.03, 0.48) | 0.19 (0.07, 0.32) | -0.26 (-0.65, 0.14) | 0 (0, 0) | 0.11 (0.03, 0.2) | .03 |
| Anti-HLD medication and/or abnormal lab(s) (TG, CHOL) (M4)/ Hyperlipidemia (M1) | -0.07 (-0.22, 0.07) | 0.18 (-0.08, 0.45) | 0.2 (-0.003, 0.41) | 0.18 (0.08, 0.29) | -0.07 (-0.23, 0.09) | 0 (0, 0) | 0.09 (0.05, 0.12) | .02 |
| Anti-HTN medications (M3)/ Systemic hypertension (M1) | 0.17 (-0.21, 0.55) | 0.76 (0.55, 0.98) | 0.12 (-0.27, 0.5) | 0.55 (0.39, 0.72) | 0.25 (0.02, 0.48) | 0.42 (0.01, 0.82) | 0.36 (0.27, 0.45) | .003 |
| Abnormally high BP (M2)/ Systemic hypertension (M1) | 0.16 (-0.3, 0.61) | 0.23 (-0.21, 0.68) | 0.18 (-0.2, 0.56) | 0.2 (-0.02, 0.42) | 0.31 (-0.04, 0.66) | -0.12 (-0.35, 0.1) | 0.17 (0.06, 0.28) | NS |
| Anti-hypertensive medications and/or abnormally high BP (M4)/ Systemic hypertension (M1) | 0.16 (-0.11, 0.44) | 0.65 (0.4, 0.9) | 0.15 (-0.14, 0.45) | 0.41 (0.26, 0.56) | 0.24 (0.08, 0.39) | 0.09 (-0.17, 0.34) | 0.27 (0.2, 0.34) | .02 |
| Vascular disorder medication (M3)/ Vascular disorder (including ICD) (M1) | 0 (0, 0) | 0.04 (-0.17, 0.24) | 0 (0, 0) | 0 (0, 0) | 0.05 (-0.01, 0.1) | 0 (0, 0) | 0.02 (0.01, 0.04) | NS |
| Kidney medication (M3)/ Kidney disorder (M1) | ** | 0 (0, 0) | ** | 0 (0, 0) | 0 (0, 0) | ** | -0.004 (-0.01, 0.003) | ** |
| Abnormally high lab(s) (CR, BUN) (M2)/ Kidney disorder (M1) | 0 (0, 0) | -0.05 (-0.13,.03) | 0 (0, 0) | 0.03 (-0.03, 0.09) | -0.03 (-0.07, 0.01) | 0 (0, 0) | 0.11 (0.02, 0.2) | .02 |

*(Continued)*

**Table 4.** (Continued)

| Determinant (M)/ Medical disorder (M1) | Kappa Statistic, (95% CI) | | | | | | | P-Value |
|---|---|---|---|---|---|---|---|---|
| | AUS N = 27 | CHN N = 34 | DEU N = 23 | IND N = 111 | ISR N = 47 | ITA N = 23 | USA N = 460 | |
| Kidney medication and/or abnormally high lab(s) (CR, BUN) (M4)/ Kidney disorder (M1) | 0 (0, 0) | -0.05 (-0.13,.03) | 0 (0, 0) | 0.03 (-0.03, 0.09) | -0.03 (-0.07, 0.01) | 0 (0, 0) | 0.11 (0.02, 0.2) | .02 |
| Obesity medication (M3)/ Obesity (M1) | 0 (0, 0) | 0 (0, 0) | -0.1 (-0.19, -0.002) | 0 (0, 0) | 0.01 (-0.24, 0.26) | 0 (0, 0) | -0.03 (-0.04, -0.01) | NS |
| BMI ≥ 30 (M2)/ Obesity (M1) | 0 (0, 0) | 0 (0, 0) | 0.35 (-0.03, 0.73) | 0 (0, 0) | 0.31 (0.03, 0.59) | 0 (0, 0) | 0.05 (0.01, 0.08) | NS |
| Obesity medication and/or BMI ≥ 30 (M4)/ Obesity (M1) | 0 (0, 0) | 0 (0, 0) | 0.26 (-0.05, 0.57) | 0 (0, 0) | 0.18 (0.02, 0.34) | 0 (0, 0) | 0.03 (0.01, 0.05) | NS |
| Physician diagnosed prior NAION (M2)+/ prior NAION in fellow eye (M1) | 0.84 (0.53, 1) | 0.37 (-0.16, 0.9) | 1 (1, 1) | 0.28 (0.08, 0.48) | 0.62 (0.35, 0.9) | 0.5 (0.01, 0.99) | 0.74 (0.67, 0.81) | .001 |
| Psychiatric medication (M3)+/ Psychiatric disorder (M1) | 0.51 (0.03, 0.99) | ** | -0.07 (-0.18, 0.04) | 1 (1, 1) | 0.45 (0.11, 0.79) | 0 (0, 0) | 0.51 (0.42, 0.61) | <.0001 |

+also used for M4.

*Determinants include laboratory measures, medications, and/or physical exams;

**Characteristics are presented as frequency (percent);

***N (%) of patients' labs and clinical exams exclude those who took medication for the disease/disorder observed.

targeted illness, acute NAION. Our report highlights that even with patients who have a high medical acumen (given the understanding and willingness to participate in a clinical trial), that medical histories, particularly when self-reported, are frequently inaccurate. Our combination method, using laboratory test results, physical exam findings and medications coupled with medical histories can be used by sponsors conducting clinical trials to determine more complete assessments of systemic disease, and possible confounding illnesses.

Precise assessment of all illnesses and comorbidities is needed for clinical trial management as well as for recommendations for each study participant based on baseline illness status. Administrative, data management, and sponsor issues notwithstanding, improved accuracy is needed for future considerations, for optimal management of participants during a study, and for improving the clinical trial results.[24] Machine learning and other types of artificial intelligence methods are currently being applied to clinical trial datasets in a post hoc manner.[25] Multiple variables are often considered in the evaluation of the trial results, but if all the variables directly associated with the target or associated disease are not included, the results of analyses could be flawed or not reproducible.[26] This also has implications for other patient management and research projects, which explore associations between EHR derived data and gene studies.

Our results show variation across countries, but there was generally under-reporting of medical disorders and poor agreement with the medications and the laboratory and physical exam findings for each medical disorder. However, for systemic hypertension, hyperlipidemia, and prior fellow eye NAION, all of which are verifiable by one or more of our determinants, the participants over-reported these conditions. Female and male participants were similar, showing good to very good concordance only for medications or exam findings for diabetes mellitus, prior fellow eye NAION, and psychiatric disease. The generally strong agreement with reporting prior fellow eye NAION and the physical exam finding, is due in part to the specific clinical trial for this disorder so participants were motivated to be screened and enrolled in the treatment trial. We did not explore the relationship with age, given that the study entry criteria limited participants to older adults. We cannot comment on the medical history reporting from EHRs across countries or sites as the trial database did not detail how often they were used by site investigators or coordinators.

This cohort study suggests that cultural issues may have a role for reporting and treating illnesses such as psychiatric disorders. It is also important to note that there are no laboratory confirmatory tests to adequately evaluate the presence of psychiatric disorders.

Our findings, drawn from data collected under controlled conditions, also have important implications for "big data" approaches utilizing healthcare data sources, which are not collected under optimal conditions. EHR reviews frequently under-report the prevalence of medical illness, and results rarely include medications or medication indications or diagnostic test results.[27–29] Comparison studies suggest that self-reported surveys yield higher prevalence rates compared with electronic medical data extraction except for severe illnesses.[30] There are few studies on the actual sensitivity and specificity of procedures used to identify diagnoses from survey or medical records,.[31] Additionally, prevalence reports have not used consistent definitions or evidence criteria.[32] We suggest that population health survey and 'big data' evaluations from healthcare system records need verifying evidence. Few reports describe the use of all the information available to describe the prevalence of disease.[27] Even the prevalence of comorbidity data for targeted diseases varies widely depending on the type of records searched.[33] Reasons for the variability include incomplete records, non-uniform recruitment criteria or stage of illness, and medical records data collection driven by patients seeking care or the severity of illness [34] These issues are not easily overcome, but including all the data in the medical records will likely improve diagnostic precision. Machine learning is currently in use to evaluate 'Big Data'[35], but artificial intelligence methods cannot overcome training on only part of the variables associated with each disease and may lead to models that incompletely identify risk factors or provide spurious correlations.[36,37] Our results suggest that such analyses should incorporate multiple types of medical information to enhance accuracy.

Although we might assume that symptomatic or clinically obvious disorders would have a higher prevalence than asymptomatic conditions [29], our study did not show this. For example, obesity, cardiac and ischemic heart disease prevalence were markedly under-reported by the medical history. However, one symptomatic medical illness, psychiatric disorder, showed excellent concordance between medical history and medication. The variability of patient reporting of symptomatic illnesses and medical history under-reporting, particularly when minimally symptomatic or asymptomatic, is known.[4] We assumed that our study participants with a cardiac disorder did not have severe disease, due to the clinical trial exclusion criteria. The under-reporting of less symptomatic medical conditions in our study participants is in concert with what is recognized when EHRs are interrogated.

Poor documentation for all aspects of medical information further limit the accuracy and increase discrepancies when comparing medical history to other methods to determine disease prevalence.[38] Using billing codes or problems lists as the gold standard for identifying chronic conditions and multi-morbidity data only has moderate to good agreement with other methods, which is in part due to variability of healthcare provider evaluations and documentation.[39] Similar issues occur during a clinical trial if the priority for data entry and monitoring is focused solely on issues that are known to be associated with the study outcomes and safety.

Our report has limitations. Our method of listing multiple indications for drugs could have overstated the disparity between the medical history information and disorder-specific medications. For example, medications used to treat ischemic heart disease could be to treat systemic hypertension or cerebrovascular and peripheral vascular disorders. Thus, we might have overestimated the prevalence of some disorders based on the medication indications. We did not consider clinical disorders for which the clinical trial did not have data for one of our verifying methods. We did not consider all issues related to obesity, and most healthcare system EHRs do not contain all of the types of data that are relevant. Medical disorders that had a low frequency in the study cohort might have too few participants to report accurately. It is also possible that potential participants might omit information that might exclude them from being enrolled in a study. Last, the study cohort was restricted by entry criteria, 50–80 years of age and stable medical illnesses. This study does not purport to report the prevalence of the medical illnesses beyond the study cohort.

Despite the study limitations, we conclude that health surveys, EHR, and clinical trial analyses need to consider all potential indicators in determining the prevalence of medical illnesses. It is possible that more complete data that includes the patient illness and findings for which a medication is prescribed will reduce potential artificially increased prevalence based on drug indication databases. Using these and additional measures should improve the accuracy of reporting.

Data access: Others who seek access to the datset used in this study have to contact Quark Pharmaceutical, Inc for a data sharing agreement. Quark Pharmaceutical, Inc allowed MJK access to the data for academic use. The binding agreement with the company prevents distribution of the data for use as the database contains proprietary information that could be used by competitor drug companies. For data Inquires danzurr@leptonpharma.com can be contacted.

## Strengths and limitations

- Prospective rigorously collected data utilized.

- Cohort of participants with acute non-arteritic anterior ischemic neuropathy for a trial that excluded some active medical illnesses.

- Use of medicine indications might lead to over estimation of medical illnesses.

## Author contributions

**Data curation:** Rachelle Morgenstern.

**Formal analysis:** Mark Kupersmith, Rachelle Morgenstern, Nathalie Jette.

**Funding acquisition:** Mark Kupersmith.

**Methodology:** Avi Reichenberg, Nathalie Jette.

**Project administration:** Mark Kupersmith.

**Supervision:** Mark Kupersmith.

**Writing – original draft:** Mark Kupersmith, Benjamin Kummer.

**Writing – review & editing:** Mark Kupersmith, Avi Reichenberg, Benjamin Kummer, Nathalie Jette.

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
