## [Decision Letter · Decision Letter 0]

25 Feb 2025

The Reliability of Medical Illness Reporting in a Randomized Clinical Trial

PONE-D-24-41310

Dear Dr. Kupersmith,

We’re pleased to inform you that your manuscript has been judged scientifically suitable for publication and will be formally accepted for publication once it meets all outstanding technical requirements.

Kind regards,

Alvaro Jose Mejia-Vergara, MD

Academic Editor

PLOS ONE

 Please include your full ethics statement in the ‘Methods’ section of your manuscript file. In your statement, please include the full name of the IRB or ethics committee who approved or waived your study, as well as whether or not you obtained informed written or verbal consent. If consent was waived for your study, please include this information in your statement as well.

Reviewers' comments:

Reviewer's Responses to Questions

**Comments to the Author**

1. Is the manuscript technically sound, and do the data support the conclusions?

Reviewer #1: Yes

2. Has the statistical analysis been performed appropriately and rigorously? 

Reviewer #1: Yes

3. Have the authors made all data underlying the findings in their manuscript fully available?

Reviewer #1: Yes

4. Is the manuscript presented in an intelligible fashion and written in standard English?

Reviewer #1: Yes

5. Review Comments to the Author

Reviewer #1: The manuscript presents a scientifically rigorous study with the research conducted meticulously, incorporating appropriate measures. The sample sizes were adequate yielding robust and reliable results. The conclusions were well-supported by the data meeting the standards. The research questions were novel and contributed in a meaningful way to the field. The methodology was described in sufficient detail for reproducibility. The data was presented clearly and support the conclusions that were summarized. Effective discussion of possible bias and limitations within your study. The manuscript was well-organized and written in a clear and concise manner.

6. PLOS authors have the option to publish the peer review history of their article (what does this mean? ). If published, this will include your full peer review and any attached files.

**Do you want your identity to be public for this peer review?** For information about this choice, including consent withdrawal, please see our Privacy Policy .

Reviewer #1: No

---

## [Editor Report · Acceptance letter]

PONE-D-24-41310

PLOS ONE

Dear Dr. Kupersmith,

I'm pleased to inform you that your manuscript has been deemed suitable for publication in PLOS ONE. Congratulations! Your manuscript is now being handed over to our production team.

Kind regards,

on behalf of

Dr. Alvaro Jose Mejia-Vergara

Academic Editor

PLOS ONE